# FlowOpt: Fast Optimization Through Whole Flow Processes for Training-Free Editing

## Abstract

The remarkable success of diffusion and flow-matching models has ignited a surge of works on adapting them at test time for controlled generation tasks. Examples range from image editing to restoration, compression and personalization. However, due to the iterative nature of the sampling process in those models, it is computationally impractical to use gradient-based optimization to directly control the image generated at the end of the process. As a result, existing methods typically resort to manipulating each timestep separately. Here we introduce FlowOpt – a zero-order (gradient-free) optimization framework that treats the entire flow process as a black box, enabling optimization through the whole sampling path without backpropagation through the model. Our method is both highly efficient and allows users to monitor the intermediate optimization results and perform early stopping if desired. We prove a sufficient condition on FlowOpt's step-size, under which convergence to the global optimum is guaranteed. We further show how to empirically estimate this upper bound so as to choose an appropriate step-size. We demonstrate the effectiveness of FlowOpt in the context of image editing, showcasing two use cases: ($i$) inversion (determining the initial noise that generates a given image), and ($ii$) directly steering the edited image to be similar to the source image while conforming to the target text prompt. In both settings, our method achieves state-of-the-art results while using roughly the same number of neural function evaluations (NFEs) as existing methods.

## 1 Introduction

Diffusion and flow matching models have emerged as powerful generative frameworks, achieving state-of-the-art (SotA) results on image, video, and audio generation (Ho et al., 2020; Song et al., 2021a; Rombach et al., 2022; Lipman et al., 2023; Liu et al., 2023; Albergo & Vanden-Eijnden, 2023). However, as opposed to their generative adversarial network (GAN) predecessors, flow models generate samples through an iterative process that often involves dozens of neural function evaluations (NFEs). This makes it challenging to adapt them at inference time for solving controlled generation tasks. Indeed, while GANs naturally lend themselves to gradient-based optimization for directly minimizing losses on the generator's output (Menon et al., 2020), in flow models this approach is computationally impractical. As a result, methods that use pre-trained flow models for controlled generation typically intervene in each step of the sampling process separately, without employing any direct supervision on the final result. This strategy is used *e.g.*, for image restoration, image editing (using inversion techniques), and image compression (Kawar et al., 2022; Tumanyan et al., 2023; Pan et al., 2023; Qi et al., 2023; Huberman-Spiegelglas et al., 2024; Hong et al., 2024; Cohen et al., 2024; Garibi et al., 2024; Manor & Michaeli, 2024; Elata et al., 2025; Wang et al., 2025; Martin et al., 2025; Deng et al., 2025; Ohayon et al., 2025; Samuel et al., 2025).

Recently, Ben-Hamu et al. (2024) demonstrated the great potential of employing optimization through the whole flow process in the context of solving inverse problems with pre-trained flow models. Unlike other methods, this approach directly controls the generated image, and thus avoids accumulation of approximation errors that can build up throughout the flow path. However, performing gradient-based optimization is not scalable to reasonably sized models and image dimensions. In fact, even with a small flow-matching model, small images ($128 \times 128$), and memory-saving techniques like gradient checkpointing, this approach takes approximately 15 minutes to run on a single input.

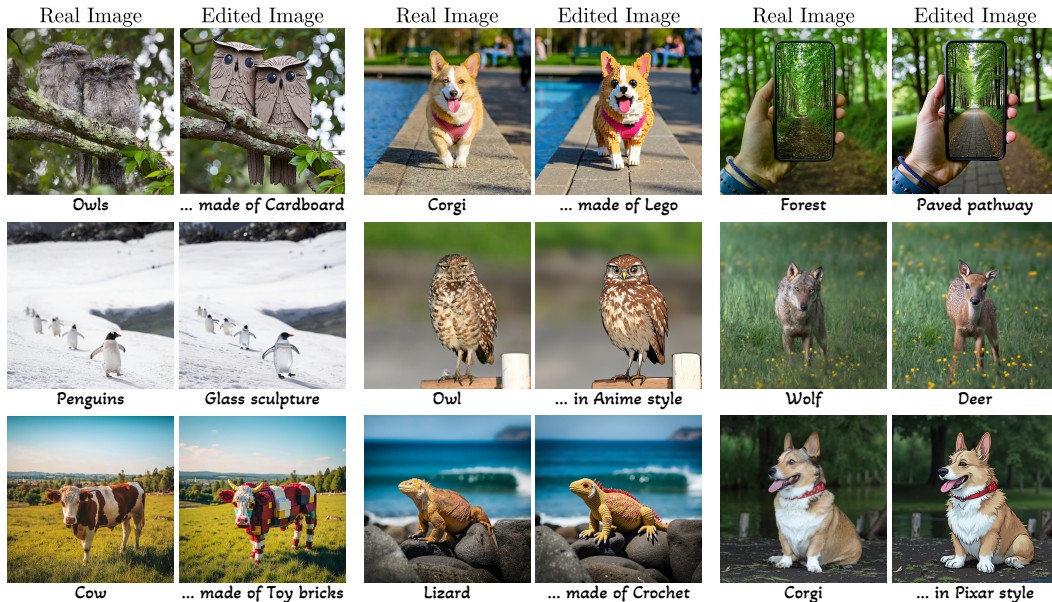

**Figure 1: FlowOpt.** We propose a zero-order (gradient-free) framework for optimization through an unrolled flow sampling process. FlowOpt can efficiently optimize losses on the target image, even when working with large models and high resolution images. We leverage our framework for text-based image editing, demonstrating state-of-the-art results on both FLUX (first and third rows) and Stable Diffusion 3 (second row). Fine details are visible upon zooming in.

In this work, we introduce FlowOpt – a zero-order (gradient-free) optimization framework for directly minimizing loss functions on the target image without backpropagating through the model. Specifically, unrolling the sampling process, a flow model can be viewed as a chain of neural networks, which we refer to as "denoisers". Our approach treats this entire chain of denoisers as a black box, and enables optimization with respect to arbitrary loss functions. Here we specifically focus on image-editing objectives. The avoidance of backpropagation enables working with large flow models and treating large images. Furthermore, it allows using a small number of flow timesteps, which is in contrast with inversion-based techniques that often require many timesteps to avoid error accumulation. Taken together, these features enable FlowOpt to achieve SotA results at a number of NFEs comparable to existing methods. Additionally, FlowOpt allows monitoring the intermediate optimization results. Thus, at the same budget of NFEs as existing methods, FlowOpt in fact provides multiple candidate edited images (one per optimization step) from which the user can choose.

Zero-order optimization has been previously used in several computer vision contexts (Tao et al., 2017; Milanfar, 2018; Chen et al., 2019; Tu et al., 2019). FlowOpt is a generalization of the method of Tao et al. (2017), with the difference that the update in each optimization step is multiplied by a step-size $\eta$ (the method of Tao et al. (2017) corresponds to FlowOpt with $\eta = 1$). As we show, this modification is of dramatic importance. Specifically, we prove a sufficient condition on $\eta$ under which convergence to the global minimum is guaranteed, and show that for popular flow models this bound is orders of magnitude smaller than 1. We demonstrate that FlowOpt indeed converges when $\eta$ is chosen smaller than the bound, and fails to converge when it significantly exceeds the bound.

We demonstrate the effectiveness of FlowOpt for both image reconstruction (inversion) and direct image editing (Fig. 1), using the FLUX-1.dev (Black Forest Labs, 2024) and Stable Diffusion 3 (SD3) (Esser et al., 2024) text-to-image (T2I) models. We show that FlowOpt provides an efficient solution to these tasks, delivering SotA performance at running times comparable to existing methods.

## 2 RELATED WORK

T2I diffusion and flow-based models (Saharia et al., 2022; Ramesh et al., 2022) generate images by steering a diffusion or flow process according to a text prompt provided by the user. Latent diffusion and flow-based variants (Rombach et al., 2022; Vahdat et al., 2021; Dao et al., 2023) follow the same principle but operate in a lower-dimensional latent space, improving computational efficiency while

preserving visual fidelity. Many methods utilize these T2I foundation models for downstream tasks like image editing in a zero-shot manner.

A common approach for performing image editing with pre-trained diffusion/flow models is to start with an inversion stage (Song et al., 2021a) (often referred to as DDIM or ODE inversion), whose goal is to extract the initial noise that would generate the input image if used in a regular sampling process. Once this initial noise is obtained, it is used for sampling a new image, by using a text prompt that describes the desired edit. However, inversion methods introduce approximation errors that accumulate across the flow timesteps, and lead to significant reconstruction inaccuracies (Mokady et al., 2023; Huberman-Spiegelglas et al., 2024).

One line of work focuses on improving the precision of ODE-inversion. Wang et al. (2025) employ a high-order Taylor expansion to more accurately approximate the nonlinear components of the flow. Deng et al. (2025) propose a solver that reuses intermediate velocity vector approximations. Yet, despite improving numerical accuracy, such methods still operate on each timestep separately and do not promote direct alignment with the given image during the inversion. Therefore, they still suffer from accumulation of errors that can degrade overall performance.

A different approach is to optimize each denoising timestep independently (Mokady et al., 2023; Pan et al., 2023; Hong et al., 2024; Garibi et al., 2024; Miyake et al., 2025; Samuel et al., 2025). For instance, Mokady et al. (2023) optimize the unconditional null prompt embedding used in classifier-free guidance (CFG) (Ho & Salimans, 2021) during the reverse process, aligning latent variables obtained through DDIM inversion. While effective, this approach requires storing all latent variables and optimized embeddings in memory, which becomes prohibitive for a large number of timesteps. Furthermore, repeated backward passes through each timestep render such methods impractical for interactive editing with large-scale models. Hong et al. (2024) propose a gradient-based inversion scheme applied independently at each timestep, however their method is computationally expensive and time-intensive, particularly for modern large-scale T2I models. Pan et al. (2023) and Garibi et al. (2024) mitigate this by introducing fixed-point iteration strategies that iteratively refine approximations of predicted states along the diffusion trajectory. However, all these methods rely on optimizing each timestep independently, ignoring the input image in each optimization step. This leads to accumulation of local approximation errors that degrade overall performance.

There exist several optimization-based methods that may superficially seem similar to FlowOpt, as they neglect the Jacobian of the denoiser and thus avoid backpropagation through the model. These include Score Distillation Sampling (SDS) (Poole et al., 2023), Delta Denoising Score (DDS) (Hertz et al., 2023), Posterior Distillation Sampling (PDS) (Koo et al., 2024), and inverse Rectified Flow Distillation Sampling (iRFDS) (Yang et al., 2025). However, these methods still optimize each timestep separately by randomly sampling a timestep in each optimization step and performing an update based on that timestep alone. This is in contrast with FlowOpt, which performs optimization through the whole chain of denoisers simultaneously.

Recently, Patel et al. (2025) proposed FlowChef, a method that initializes the sampling process from white Gaussian noise, and then performs zero-order optimization at each denoising timestep separately. Unlike FlowOpt, this method does not treat the entire flow process as a black box. A detailed comparison between the two methods is provided in App. K.

Finally, Ben-Hamu et al. (2024) proposed D-Flow, a method that like FlowOpt, optimizes across the entire generative process. However, their framework relies on gradient-based optimization and requires repeated backpropagation through the entire chain of denoisers. This makes the method computationally intensive and impractical for high-resolution, real-world applications – precisely the setting we aim to address with FlowOpt.

## 3 PRELIMINARIES AND NOTATION

Probability flow ODE (Song et al., 2021b) and flow-matching models (Lipman et al., 2023; Liu et al., 2023; Albergo & Vanden-Eijnden, 2023) generate images by numerically solving an ODE over a time parameter $t$. Focusing for simplicity on the flow-matching formalism, the ODE takes the form

$$d\boldsymbol{z}_t = \boldsymbol{v}_t(\boldsymbol{z}_t, c)\, dt, \quad t \in [0, 1]. \tag{1}$$

This ODE is designed such that when initialized at $t = 1$ with a sample from some source distribution (usually taken to be an isotropic Gaussian), $\boldsymbol{z}_1 \sim \pi_1$, and run backwards in time until $t = 0$, it yields

**Figure 2: A whole flow process as a black box.** We encapsulate the flow process as a black box function $f$, which receives an initial noise $z_1$ and text conditioning $c$, and outputs a clean sample $z_0$. Each internal step within the black box is given by $\psi_t(z_t, c) = z_t + v_t(z_t, c)\Delta t$, where $v_t$ is the text-conditioned velocity predicting network.

a sample from a desired target distribution (*e.g.* the distribution of natural images), $z_0 \sim \pi_0$. The function $v_t(\cdot, \cdot)$ is a time dependent vector field that optionally accepts a condition $c$ (*e.g.*, a text prompt) in its second argument. In practice, this velocity field is implemented by a neural network, which we refer to as "denoiser", and the ODE is discretized and solved numerically as

$$z_{t+\Delta t} = z_t + v_t(z_t, c)\,\Delta t, \tag{2}$$

where $\Delta t$ is the (negative) discretization step.

Unrolling Eq. (2), the sample $z_0$ generated at the end of the flow process can be written as a function of the initial noise $z_1$, namely $z_0 = f(z_1, c)$. This function is given by

$$f(z_1, c) = z_1 + \sum_i v_{t_i}(z_{t_i}, c)\,\Delta t, \tag{3}$$

where $t_i = 1 + i\,\Delta t$ (see Fig. 2). For notational simplicity, we henceforth omit the condition $c$ whenever it is clear from the context. Furthermore, we sometimes use $f(\cdot)$ to denote the mapping from some intermediate timestep $t < 1$ to timestep $t = 0$. Our method treats the function $f(\cdot)$ as a black box in the sense that it can be evaluated but its Jacobian cannot be computed.

Commonly, the flow process is defined in the latent space of an encoder $\mathcal{E}(\cdot)$, so that the final image is obtained by passing the generated sample $z_0$ through the corresponding decoder $\mathcal{D}(\cdot)$.

## 4 METHOD

Given a source image $y$, a text prompt $c_{\text{src}}$ describing it, and a target text prompt $c_{\text{tar}}$ describing a desired edit, our goal is to generate an edited image $y_{\text{edit}}$ that conforms to $c_{\text{tar}}$ while being as similar as possible to $y$. Like previous approaches, we rely on a pre-trained flow model. However, in contrast to existing methods we propose to achieve this by directly optimizing over the vector $z_t$ at some timestep $t$ (usually taken to be 1), such that the image $z_0$ at the end of the flow process is close to $y$.

Formalizing this mathematically, we are interested in $z_t^* = \arg\min_{z_t} \mathcal{L}(f(z_t, c), y)$, where $\mathcal{L}$ is some dissimilarity measure. Let us focus on the $L^2$ loss (see App. E for other losses). In this case,

$$z_t^* = \arg\min_{z_t} \frac{1}{2}\|f(z_t, c) - y\|^2. \tag{4}$$

This optimization problem can be used in two distinct ways. (*i*) **Inversion:** setting $c = c_{\text{src}}$ in Eq. (4) leads to a $z_t^*$ that reconstructs the input image with the source prompt. (*ii*) **Direct editing:** setting $c = c_{\text{tar}}$ in Eq. (4) leads to a $z_t^*$ that directly approximates the input image with the target prompt. In both cases, once $z_t^*$ is obtained, it can be used to generate the edited image by performing sampling with the target prompt, $y_{\text{edit}} = f(z_t^*, c_{\text{tar}})$.

Using gradient descent to solve Eq. (4) would lead to the iterations

$$z_t^{(i+1)} \leftarrow z_t^{(i)} - \eta\, \boldsymbol{J}(z_t^{(i)})^\top \left( f(z_t^{(i)}) - y \right), \tag{5}$$

where $\eta$ is the step size and $\boldsymbol{J}(z_t^{(i)})$ is the Jacobian of $f(\cdot)$ with respect to $z_t^{(i)}$. However, as mentioned above, backpropagation through whole flow processes is computationally impractical.

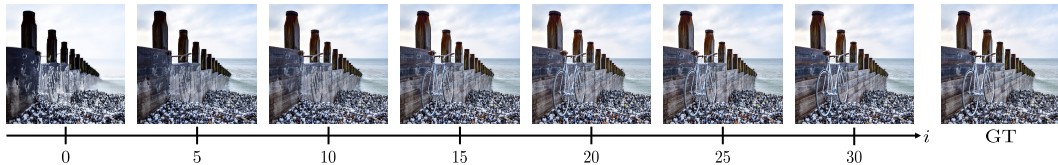

**Figure 3: Image inversion with FlowOpt.** Intermediate samples $z_0^{(i)} = f(z_t^{(i)}, c)$ attained during our zero-order optimization through a chain of 10 denoising steps (FLUX) for the task of reconstruction (inversion), *i.e.*, with $c = c_{\text{src}}$. Notice the missing details in the early steps, such as the bicycle and the horizon. As the iterations progress, the reconstruction converges to the ground truth image.

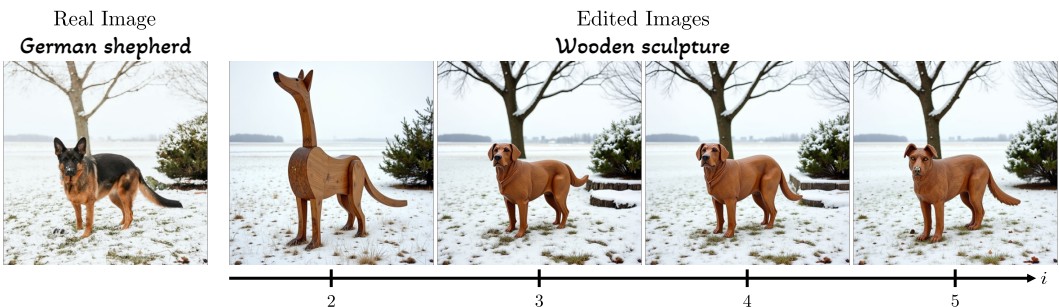

**Figure 4: Direct image editing with FlowOpt.** Intermediate samples $z_0^{(i)} = f(z_t^{(i)}, c)$ attained during our zero-order optimization through a chain of 15 denoising steps (FLUX) for direct image editing, *i.e.*, with $c = c_{\text{tar}}$. Notice the misalignment in the dog's body structure in the first iterations.

Therefore, as an alternative, here we propose to simply ignore the Jacobian. This leads to the zero-order (gradient-free) iterations

$$z_t^{(i+1)} \leftarrow z_t^{(i)} - \eta \left( f(z_t^{(i)}) - y \right). \tag{6}$$

Figure 3 demonstrates the progression of those iterates when used for inversion (with the source prompt). Figure 4 demonstrates the progression of the iterates when used for direct editing (with the target prompt). Algorithm 1 summarizes the proposed method.

Before providing a theoretical convergence guarantee, two comments are in place. First, when $\eta = 1$, Eq. (6) degenerates to the method of Tao et al. (2017). However, as we show below, $\eta$ is of crucial importance, as the maximal step size allowing convergence is much smaller than 1 for modern flow-matching models. Second, it is insightful to note that for flow-matching models, Eq. (6) is equivalent to using gradient descent with step-size $\eta$ while applying the `stop-grad` operator on the output of the velocity prediction network. Similarly, for probability flow ODE models (Song et al., 2021b), (a.k.a. DDIM (Song et al., 2021a)), Eq. (6) is equivalent to using gradient descent with step size $\sqrt{\alpha_T}\eta$ while applying `stop-grad` on the noise prediction network (following the notation of Song et al. (2021a)). The derivations of those observations are provided in App. G.

The iterations of Eq. (6) can be written as $z_t^{(i+1)} = g(z_t^{(i)})$, where $g(u) \triangleq u - \eta(f(u) - y)$. By the Banach fixed-point theorem, if $g(\cdot)$ is a contractive mapping[1] then there exists a unique point satisfying $z_t^* = g(z_t^*)$, and thus $f(z_t^*) = y$. Furthermore, in this case the iterations converge to this unique solution. This fact can be used to obtain a sufficient condition on the step size $\eta$ under which the iterations are guaranteed to converge to the global minimum (see proof in App. F).

**Theorem 1.** *Assume that* $\inf\limits_{u_1 \neq u_2} \frac{\langle u_1 - u_2, f(u_1) - f(u_2) \rangle}{\|u_1 - u_2\| \|f(u_1) - f(u_2)\|} > 0$ *and* $\sup\limits_{u_1, u_2} \frac{\langle u_1 - u_2, f(u_1) - f(u_2) \rangle}{\|f(u_1) - f(u_2)\|^2} < \infty.$
*If the step size $\eta$ satisfies*

$$0 < \eta < 2 \inf_{u_1, u_2} \frac{\langle u_1 - u_2, f(u_1) - f(u_2) \rangle}{\left\| f(u_1) - f(u_2) \right\|^2} \tag{7}$$

*then there is a unique $z_t^*$ satisfying $f(z_t^*) = y$ and the iterations of Eq. (6) converge to this $z_t^*$.*

---

[1]$g(\cdot)$ is a contractive mapping if it satisfies $\|g(u_1) - g(u_2)\| \leq \gamma \|u_1 - u_2\|$ for some $\gamma < 1$ and all $u_1, u_2$.

---

**Algorithm 1:** Flow Zero-Order Optimization (FlowOpt)

---

**Require:** step size $\eta$, number of iterations $N$, condition $c$, input image $\boldsymbol{y}$

**Initialization:** $\boldsymbol{z}_t^{(0)} \in \mathbb{R}^d$

**for** $i \leftarrow 0, \ldots, N-1$ **do**

$\quad$ $\boldsymbol{z}_0^{(i)} = f(\boldsymbol{z}_t^{(i)}, c)$

$\quad$ $\boldsymbol{z}_t^{(i+1)} \leftarrow \boldsymbol{z}_t^{(i)} - \eta(\boldsymbol{z}_0^{(i)} - \boldsymbol{y})$

$\boldsymbol{z}_0^{(N)} = f(\boldsymbol{z}_t^{(N)}, c)$

Return $\{\boldsymbol{z}_0^{(i)}\}_{i=0}^N$

---

**Table 1: Step sizes guaranteeing convergence**. Column 2 shows the estimated sufficient condition of Eq. (7) and column 3 reports the step size we chose for each model (see App. F for details).

| Model | Sufficient condition (Eq. (7)) | Our chosen step size |
|---|---|---|
| FLUX | $\eta < 2.70 \cdot 10^{-3}$ | $\eta = 2.5 \cdot 10^{-3}$ |
| SD3 | $\eta < 1.67 \cdot 10^{-2}$ | $\eta = 1.0 \cdot 10^{-2}$ |

The bound in Eq. (7) depends only on the flow model $f(\cdot)$. It can thus be computed once for each model in order to choose the step size. In App. F we approximate this upper bound for the FLUX and SD3 models by drawing many pairs of samples $\boldsymbol{u}_1, \boldsymbol{u}_2$. As we show, the right-hand side of Eq. (7) is smallest when $\|\boldsymbol{u}_1 - \boldsymbol{u}_2\|$ is small. Tab. 1 shows the bounds estimated for the two models, and the step sizes we chose for our experiments.

As can be seen, the bounds in Tab. 1 are significantly smaller than 1, suggesting that the method of Tao et al. (2017) is inapplicable in our setting. Indeed, Fig. 5 shows the reconstruction error along the iterations for several choices of $\eta$ when used for inversion with SD3 (results for FLUX are presented in App. F). When setting $\eta = 10^{-2}$, which is below the bound of $1.67 \cdot 10^{-2}$, the iterations converge. However, when using larger step sizes, like $4 \cdot 10^{-2}$ or $5 \cdot 10^{-2}$, the iterations fail to converge. The setting of this experiment is as in Sec. 5.1. For additional convergence results with other image dimensions, please see App. J.

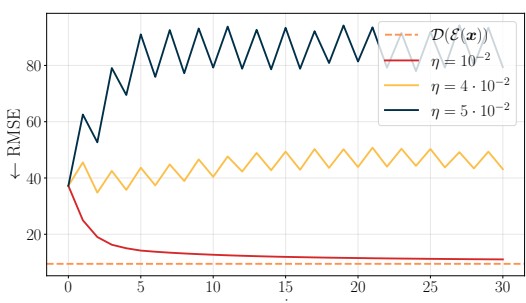

**Figure 5: Convergence analysis.** The plot shows RMSE in pixel space vs. number of iterations for the task of inversion, averaged over a dataset. The step size we use (red) satisfies the sufficient condition of Eq. (7) and thus leads to convergence. Step sizes that are $4\times$ and $5\times$ larger (yellow and black) do not satisfy the condition and do not lead to convergence. The dashed orange line is the minimal RMSE achievable in this setting. It corresponds to passing images through the encoder and decoder.

## 5 EXPERIMENTS

We compare FlowOpt against competing methods on two tasks: image reconstruction (inversion) and text-based image editing. We show results with FLUX-1.dev in the main text and with SD3 in App. D. We use the step sizes reported in Tab. 1 and initialize our algorithm with the UniInv (Jiao et al., 2025) inversion method (see App. C for details). All images are of dimensions $1024 \times 1024$.

### 5.1 IMAGE RECONSTRUCTION (INVERSION)

For inversion, we use $c = c_{\text{src}}$ in Eq. (4), setting it to a text prompt describing the source image. We set the number of flow steps in FLUX (number of denoisers) to $T = 10$ and evaluate the reconstruction error for various numbers of NFEs by varying the number of FlowOpt iterations $N$. Specifically, we

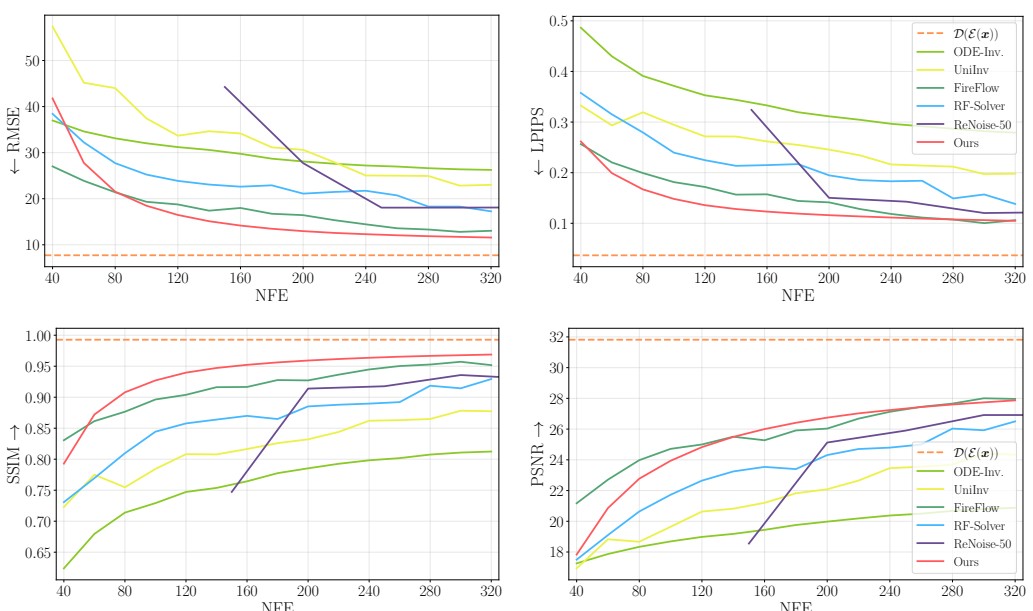

**Figure 6: Reconstruction accuracy vs. NFEs for inversion**. The plots depict pixel-space RMSE, LPIPS, SSIM, and PSNR as a function of the number of NFEs for several inversion methods. The dashed bound corresponds to passing the images through the encoder and decoder. FlowOpt achieves favorable reconstruction quality under 240 NFEs, which is the regime of practical interest.

have NFE $= T(N + 2)$, as $T$ NFEs are used for the initialization, $NT$ NFEs for the optimization process, and $T$ NFEs for the final sampling process.

We randomly choose 100 real images from the DIV2K dataset (Agustsson & Timofte, 2017), and resize and center-crop them to dimension $1024 \times 1024$. For the source prompts, we caption each image with BLIP (Li et al., 2022) and then manually refine the prompt.

We compare FlowOpt to several inversion methods: naive ODE Inversion, RF-Solver (Wang et al., 2025), FireFlow (Deng et al., 2025), UniInv (Jiao et al., 2025), and ReNoise (Garibi et al., 2024). We use the official implementations of all methods except for ODE Inversion and ReNoise (that lacks an implementation for flow models), which we implemented by ourselves. To ensure a fair comparison, we set the number of timesteps for each method such that the total NFE count is the same for all methods. Specifically, for FireFlow and UniInv, which use a single forward pass per timestep, we set $T = \frac{\text{NFE}}{2}$. For RF-Solver, which uses two forward passes per timestep for inversion and two for sampling, we set $T = \frac{\text{NFE}}{4}$. For ReNoise, we used $T = 50$ and set the number of ReNoise steps so as to achieve the desired NFE count. We note that we evaluated ReNoise with various hyperparameter settings and chose the one that achieved the best results.

Figure 6 shows the reconstruction accuracy achieved by all methods as a function of the NFEs. The figure reports pixel-space RMSE, PSNR, SSIM (Wang et al., 2004), and LPIPS (Zhang et al., 2018). As can be seen, FlowOpt achieves the best reconstruction results over a wide range of NFE counts. In App. B we show that the same trend is obtained with empty text prompts, both with the CFG parameter of FLUX set to 0 and with it set to 1 (these options differ as FLUX is a distilled model).

## 5.2 IMAGE EDITING

Accurate inversion does not necessarily lead to good editing results. Indeed, even for synthetic images, for which the initial noise map is known, plain editing-by-inversion leads to unsatisfactory results (Kulikov et al., 2025; Huberman-Spiegelglas et al., 2024) (see App. I for further discussion). Accordingly, for the task of editing we employ our direct optimization approach, where the target text prompt $c = c_{\text{tar}}$ is used in Eq. (4). In this case, we do not necessarily want a large number of iterations, to avoid getting too close to the original image. We therefore use $N \in \{2, 3, 4, 5\}$.

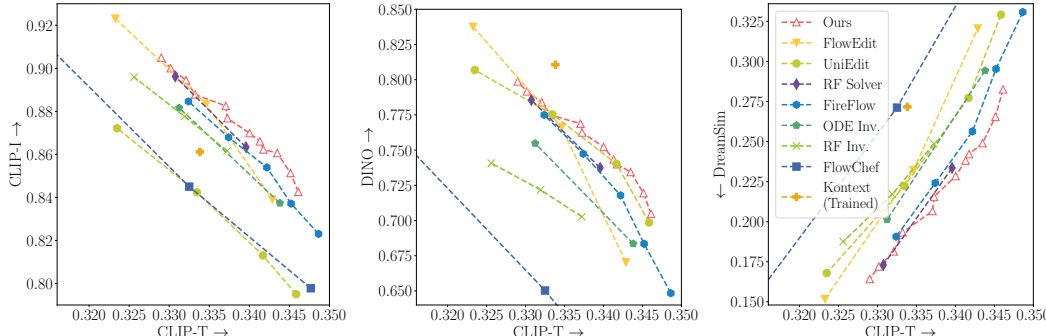

**Figure 7: Editing quantitative comparisons**. Semantic preservation of different editing methods evaluated using CLIP-Image, DINOv3 and DreamSim as functions of text adherence, measured by CLIP-Text. Connected markers represent different set of hyperparameters (see App. B). Our method achieves the most favorable balance between semantic preservation and text adherence.

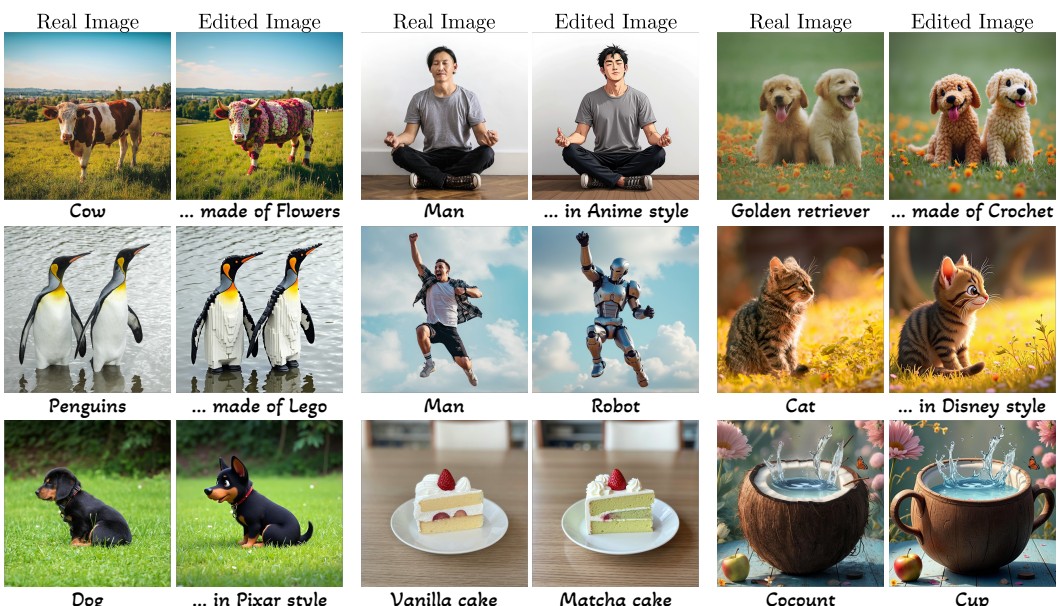

**Figure 8: FlowOpt editing results**. Our method successfully preserves the object's semantics and structure, as well as the background details, all the while loyally adhering to the target text prompt. Fine details are visible upon zooming in.

We set the number of flow steps to $T = 15$ and perform the optimization on the latent vector at timestep $n_{max} \in \{14, 13, 12\}$ (corresponding to $t$ in Eq. (4)). The total number of NFEs is given by NFE $= n_{max}(N + 2)$. We use the default CFG of 3.5. All visual results in the paper were obtained with $n_{max} = 13$, except for Fig. 1, whose hyperparameters are provided in App. H.

We evaluate all methods on the dataset of Kulikov et al. (2025), which we enriched with additional images and editing prompts. In total, our dataset consists of 90 real images of dimensions $1024 \times 1024$ from the DIV2K dataset and from royalty free online sources (Pexels, 2025; PxHere, 2025). Each image was captioned by LLaVA-1.5 (Liu et al., 2024) and manually refined. For each image, we manually created target editing prompts. Overall, this led to about 400 text-image pairs.

We compare our method against all aforementioned methods, in addition to FlowEdit (Kulikov et al., 2025), FlowChef (Patel et al., 2025) and RF-Inversion (Rout et al., 2025). These three methods were excluded from the inversion experiments of Sec. 5.1 as they do not use inversion in the regular sense (FlowEdit is inversion-free and RF-Inversion and FlowChef explicitly incorporate the source image into the denoising process). For ODE Inversion, we apply the same number of NFEs as our

method. For other methods, we use the hyperparameters reported in the papers or in the official implementations. We performed a hyperparameter search for all methods that provided more than a single set of hyperparameters. Additional details and the final hyperparameters chosen for each method are provided in App. B. In addition to this set of zero-shot methods, we further compare to FLUX Kontext (Black Forest Labs et al., 2025), a trained text-based editing model.

Figures 1, 8 and S1 showcase the diverse editing capabilities of our method, including object replacement, style changes, and texture editing. FlowOpt achieves high quality, text adherent edits that also remain loyal to the source image semantics. Figure 10 presents qualitative comparisons between FlowOpt and other methods. As can be observed, our edits maintain superior alignment with the source image's structure while simultaneously adhering to the target text. For example, when turning the horse into a zebra (first row), FlowOpt successfully preserves the leg positions. Note that FLUX Kontext is a trained model; therefore, its capacity for changing the color palette of the source image is larger. For additional comparisons, see App. B.

Figure 7 presents a numerical evaluation of the results obtained for various hyperparameters. We use cosine similarity on CLIP image and text embeddings (Radford et al., 2021) to measure adherence to the original image and to the target text prompt, respectively. For image adherence, we also use cosine similarity between DINOv3 embeddings (Caron et al., 2021; Siméoni et al., 2025), as well as DreamSim (Fu et al., 2023). As can be seen, our method achieves the best tradeoff between text adherence and structure preservation.

Additionally, we evaluate our method via a user study, in which each participant was shown the reference image, an edit instruction, and two editing results – one from our method and another from a competing method. The order of the two editing results was random. We compared our method to FireFlow, FlowEdit and RF Solver, which achieve the most comparable results to FlowOpt in terms of CLIP-Text and CLIP-Image measures. Users were asked 3 two-alternative forced questions to select their preferred editing result: ($i$) visual fidelity between the reference image and the edited result, ($ii$) text alignment between the edited instruction and the edited result, and ($iii$) overall. We collect 60 user responses, covering a sample size of 600 for each question asked for each method. The results are reported in Fig. 9, where the error bars correspond to $95\%$ confidence intervals, computed using the Wilson method (Wilson, 1927). These results support the quantitative results in Fig. 7. Additional details on the user study are provided in App. B.2.3.

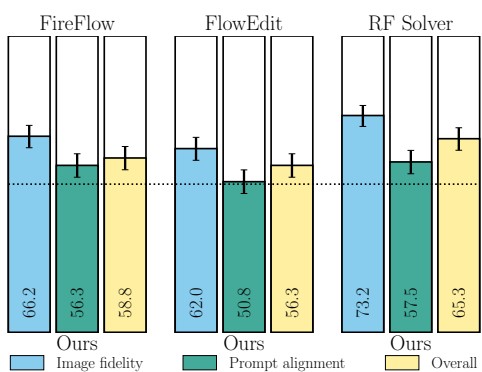

**Figure 9: Human perceptual study.** The bar plots report the percentages of users that preferred our method over competing methods in ($i$) image fidelity, ($ii$) text alignment, and ($iii$) overall. Error bars show $95\%$ confidence intervals.

## 6 CONCLUSIONS

We presented a zero-order (gradient-free) framework that allows efficient optimization over the initial noise in a flow process while minimizing a loss over the sample generated at the end of the process. We demonstrated the effectiveness of our approach for performing image editing using pre-trained flow models. In particular, extensive comparisons showed that our FlowOpt method achieves SotA performance on both image reconstruction and editing. We note that, similarly to other training-free editing methods, our approach still encounters difficulties in certain settings, like modifying large regions of the image (see App. L). However, taking a broader perspective, we believe that our zero-order framework opens the door for exploiting pre-trained flow-models in diverse applications (*e.g.*, restoration, compression, and personalization) and for diverse modalities (*e.g.*, image, video, and audio). We leave those extensions for future work.

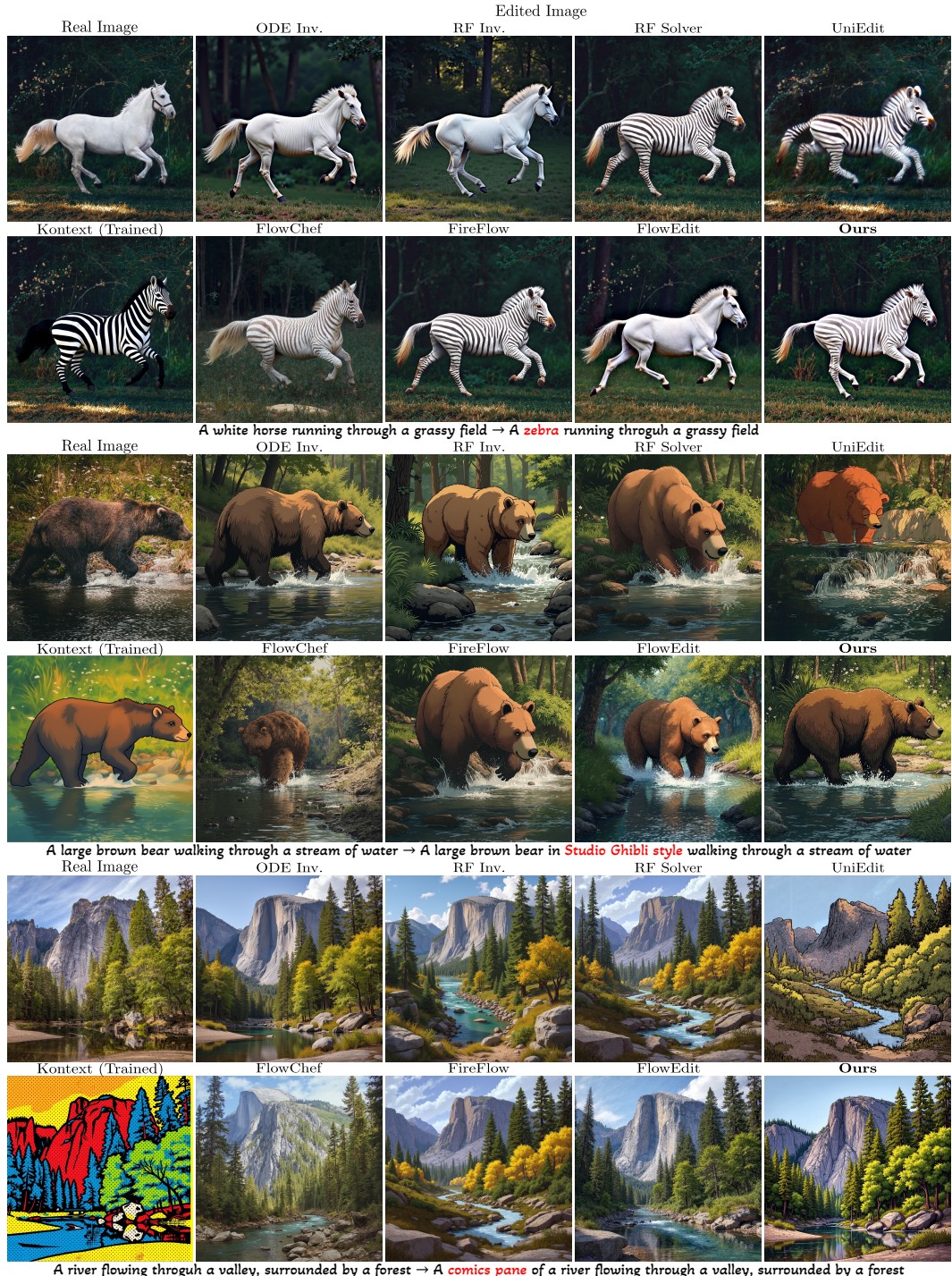

**Figure 10: Qualitative comparisons**. FlowOpt is the only method to consistently adhere both to target text prompt, and to the original image. Fine details are visible upon zooming in. For instance, the back legs of the zebra in the first row, the posture of the bear in the second row, and the structure of the scene in the last row.

ETHICS STATEMENT

This work builds upon pre-trained generative models, and thus inherits the broader ethical considerations associated with their use. Such models may reflect or amplify societal biases present in

the training data, and their outputs could be misinterpreted or misused in sensitive applications. In addition, our approach involves large-scale flow matching models, which carry the potential risk of being repurposed for harmful or malicious purposes. We emphasize that our contributions are intended solely for advancing research in generative modeling.

## REPRODUCIBILITY STATEMENT

We refer to our code repository at https://anonymous.4open.science/r/FlowOpt/. The repository includes the required scripts for running the proposed approach both for image inversion and image editing, for FLUX and SD3.

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
