# OpenReview forum: "FlowOpt: Fast Optimization Through Whole Flow Processes for Training-Free Editing"
_ICLR.cc/2026/Conference — Submitted to ICLR 2026_

### Official Review · Reviewer_NSam · 2025-10-25

**Soundness:** 3
**Presentation:** 3
**Contribution:** 4
**Rating:** 4
**Confidence:** 4

**Summary:**

The paper presents a method for image inversion and editing with flow models. The idea is to optimize a $z_t$ (typically $z_T$) to reconstruct the input image. Since it is not feasible to propagate gradients through the entire denoising process, the optimization update omits the Jacobian term. So the update step becomes $z_0^{(i)} - y$, where $z_0^{(i)}$ is the image generated from the current state of the optimization, and $y$ is the input image.
The optimization uses a small learning rate, and the paper shows that if the lr is not small enough, this process does not converge.

**Strengths:**

- The method is novel, and it is initially surprising that it works. The authors provide an analysis and theoretical justification (but I do have concerns regarding the theoretical part, see weaknesses section).
- The method itself is simple, and the paper presentation is clear.
- The authors performed extensive evaluations against competing methods and the results are plausible (but I do have concerns here, see weaknesses section).
- The limitations of the method are clearly discussed in the Appendix.
- The method's results seem to adhere to the provided edit while staying well aligned with the original image in cases where competing methods fail.

**Weaknesses:**

### Major Concerns
1. The method requires a relatively large number of NFEs in order to provide an advantage over existing methods (e.g., FireFlow and UniInv) in reconstruction.

2. The authors present a theorem that guarantees the method's convergence under certain assumptions, however why and if these assumptions hold in practice is not clear. In addition, I think that the proof itself in Appendix F is potentially flawed, as explained next.
Even assuming the condition holds, for the proof to hold we need to show that there exists some fixed $\kappa > 0$ such that the range in Eq. S8 exists. Otherwise, the limit argument is invalid for this claim.
Furthermore, there exist many functions for which the condition holds, yet for any fixed $\kappa$ the range doesn't exist. Examples include $\tanh(x)$ where the supremum of $u_1$ and $u_2$ in the expression of $\eta_1$ is infinite and $x^3$ where the infimum of $u_1$ and $u_2$ in the expression of $\eta_2$ is $0$. Both functions satisfy the required condition with $\beta = 1$.

3. While the method compares with relevant inversion-based editing methods, there are also other approaches for text-based image editing, and it is not clear that the general framework (inversion + denoising with a different prompt) is the most effective one. For example, the method is not compared with Flux Kontext or Qwen Image Edit which are the SOTA text-based image editing models.

### Minor Concerns
- Assuming that the Theorem holds, from the results in Appendix C it seems that the convergence is very slow, and in practice the initialization is crucial for the success of the method. It would be interesting to analyze convergence and performance when using other initializations, such as random noise or an interpolation between random noise and the final image.
- Analysis of performance on few-step models is missing, even though they are potentially strong candidates to benefit from this method.
- The method seems to support only appearance changes.
- Why ReNoise is not included in the editing results? And no visual results of reconstruction are provided.
- Showing other applications for this optimization framework will strengthen the paper.

### Final Note
Despite these weaknesses, I find the paper overall good. I would be willing to raise my score if the authors address the issues related to the convergence claims and provide a more thorough discussion of the origins of the method's limitations.

**Questions:**

I would like to see more experiments that empirically support the claim of convergence to a unique solution from different initial conditions. If these cannot be provided, I would suggest the convergence guarantee claims to be removed from the paper.

Methods that involve noise optimization, even gradient-free ones, can often produce inverted latents that don't exhibit properties of typical high dimensional Gaussian samples. Such properties may limit the editability of images generated by these latents (see e.g ReNoise, where the authors try to tackle this issue with regularization during optimization). I would like to see an analysis of the properties of the inverted latents found by this method, which may explain some limitations in editability, and perhaps hint towards a future solution for these limitations.

The limitation for pose editing as presented in figure S16 is counter-intuitive. I would expect that using a larger number of optimization steps would make the edited image deviate less from the original image (as is seen in Figure 4 and Figure S17), and not the other way around.

---

> ### Author Response · Authors · 2025-11-20
>
> **Large number of NFEs**
>
> We would like to kindly draw the reviewer’s attention that the number of NFEs reported in Figure 6 is both for the inversion and for the sampling. So, for example, when using 50 timesteps (a common setting for FLUX), both FireFlow and UniInv require 100 NFEs - 50 for the inversion and 50 for the sampling. As can be seen in Figure 6, at 100 NFEs our method significantly outperforms UniInv in all four metrics, and it outperforms FireFlow in three out of the four metrics.
>
> **Theorem assumptions and proof**
>
> We thank the reviewer for noting this important point. We have refined the Theorem’s assumption and its proof to account for the edge cases pointed by the reviewer.
> As the reviewer noted, when the supremum in Equation S10 is not bounded, the proof does not hold. Accordingly, we have added that assumption to the Theorem. Specifically, we added the condition that $\\sup\_{\\boldsymbol{u}\_1 \\neq \\boldsymbol{u}\_2} \\frac{ \\left\\langle \\boldsymbol{u}\_1 - \\boldsymbol{u}\_2, f(\\boldsymbol{u}\_1) - f(\\boldsymbol{u}\_2) \\right\\rangle }{\\| f(\\boldsymbol{u}\_1) - f(\\boldsymbol{u}\_2) \\| ^2} < \\infty$. To empirically validate the Theorem’s assumptions, we have added Appendix F.3 and Figure S19, where indeed we can see that these assumptions hold. Specifically, in Figure S19 we can see that the supremum is achieved for farther apart $\\boldsymbol{u}\_1$ and $\\boldsymbol{u}\_2$, and its value is finite, small, and the curve in this region is concave, i.e. it reaches a plateau.
> Additionally, in Figure S19 we can see that the assumption $\\inf\_{\boldsymbol{u}\_1 \\neq \\boldsymbol{u}\_2} \\frac{ \\left\\langle \\boldsymbol{u}\_1 - \\boldsymbol{u}\_2, f(\\boldsymbol{u}\_1) - f(\\boldsymbol{u}\_2) \\right\\rangle }{\\| \\boldsymbol{u}\_1 - \\boldsymbol{u}\_2\\| \\| f(\\boldsymbol{u}\_1) - f(\\boldsymbol{u}\_2) \\|} > 0$ holds – the infimum is achieved for closer $\\boldsymbol{u}\_1$ and $\\boldsymbol{u}\_2$, and its value is strictly positive, finite, and bounded.
> Moreover, as can be seen in Figure S17, the condition $\\inf\_{\\boldsymbol{u}\_1 \\neq \\boldsymbol{u}\_2} \\frac{ \\left\\langle \\boldsymbol{u}\_1 - \\boldsymbol{u}\_2, f(\\boldsymbol{u}\_1) - f(\\boldsymbol{u}\_2) \\right\\rangle }{ \\| f(\\boldsymbol{u}\_1) - f(\\boldsymbol{u}\_2) \\| ^2 } > 0$ also holds – for farther apart $\\boldsymbol{u}\_1$ and $\\boldsymbol{u}\_2$ the left hand side is larger than when they are closer, and it is finite and positive.
> Therefore, in Equations S9 and S10 for each constant $\\kappa$, no matter how small it is, we can find a domain that satisfies Equation S8.
>
> **Comparison to non-zero-shot editing methods**
>
> We agree with the reviewer that in light of the recent advancements in the field, it should not be taken for granted that zero-shot methods are advantageous over trained image editing models. Following the reviewer’s suggestion, we added quantitative and qualitative comparisons to FLUX Kontext (Figures 7, 10, S4 and S6, and Appendix B.2.4 in the revised manuscript). As can be seen, most state-of-the-art zero-shot methods, including ours, outperform FLUX Kontext in terms similarity to the original image (measured via CLIP-Image or DreamSim) for the level of text adherence that it achieves (measured via CLIP-Text). This seems to be rooted in the fact that there are many cases in which FLUX Kontext does not know how to perform the desired edit, and thus leaves the source image unmodified. This hurts the average text adherence score and improves the image adherence score, but overall draws it farther from the Pareto front on which the best zero-shot methods reside.
>
> **Other initializations**
>
> Indeed, FlowOpt theoretically converges geometrically from any initialization, but when starting with a poor initialization many iterations may be required for reaching a small loss. To illustrate this, we added a comparison to random noise initialization (see Appendix C). As can be seen, FlowOpt converges also with this initialization, but the convergence is slower.
>
> **Few step models**
>
> That’s an interesting point. Note that we worked with a rather small number of timesteps - we demonstrated inversion using only 10 timesteps. We get good results in this setting, even though regular sampling using 10 timesteps is suboptimal with FLUX and SD3. It is certainly possible that we can get even better results with a model that is inherently trained to work well with a few timesteps. However, to the best of our knowledge, few-step models still do not outperform the leading flow models that require many steps. We therefore leave this direction for future exploration.

---

> ### Author Response · Authors · 2025-11-20
>
> **ReNoise is not included in the editing results**
>
> ReNoise is a method which was originally developed for diffusion models, and the hyperparemeters provided in their paper and official implementation are inapplicable to flow models. The conversion between diffusion models and flow models (like FLUX and SD3) is not trivial. Despite significant attempts, we could not find a good set of hyper-parameters for the editing task. We therefore chose not to include this method in the editing evaluations.
>
> **No visual results of reconstructions**
>
> FlowOpt achieves a high PSNR in the image reconstruction task and therefore the reconstructed images are barely distinguishable from the source images. This is why we opted not to show them.
>
> **Analysis of the properties of the inverted latents**
>
> Kindly note that as opposed to many of the other methods to which we compare, we do not use inversion for performing editing. Specifically, in the editing-by-inversion paradigm the first step is to find an initial noise that reconstructs the source image with the source prompt (this is the inversion stage). In contrast, when we perform editing, we optimize for reconstruction of the source image with the target (rather than source) prompt. This directly leads to an image that is similar to the source image but also matches the target text prompt. And in this case, there is no reason for the latent to look like a sample of white Gaussian noise.
>
> **Limitation in the first Figure in the limitations section**
>
> This is a good catch. It may indeed seem that the image starts deviating from the source in the last iteration. However, this is just a small “jump” in the optimization path. We added 3 more iterations to what is now Figure S22. These iterations show that as the optimization progresses, the edited image indeed becomes more similar to the source image (and thus struggles to adhere to the target prompt).

---

### Official Review · Reviewer_j7YL · 2025-10-31

**Soundness:** 3
**Presentation:** 3
**Contribution:** 3
**Rating:** 4
**Confidence:** 3

**Summary:**

This paper addresses the task of editing images (and potentially other generative tasks)  using pre-trained flow/diffusion models in a gradient-free manner. The key idea being, treating the entire sampling process as a "black box" instead of tweaking each sampling step individually (which is the case with many existing approaches) and using a zero-order optimisation approach.

**Strengths:**

The paper is very well written.
1. This paper presents a clean idea of optimising the whole process rather than per-timestep manipulation.
2. The paper also presents a theoretical contribution: i.e., a sufficient condition on the step-size for convergence of the opimizer in this setting.
3. The edits looks visually appealing and demonstrate a good tradeoff between fidelity and edit strength.

**Weaknesses:**

1. Although the paper compares methods quantitatively and qualitatively, a user study is missing.

2. Paper doesn't really discuss how the Zero-order method performs with increase/decrease in dimension since zero-order methods may suffer from bad convergence with increase in dimension.

**Questions:**

1. Why have the authors not compared against a gradient-based inversion baseline?

---

> ### Author Response · Authors · 2025-11-20
>
> **User study**
>
> We thank the reviewer for this suggestion. To support the quantitative evaluation we reported in the initial submission (with CLIP-Image, CLIP-Text, DINOv3 and DreamSim), we now conducted a user study, which is reported in Section 5.2 in the revised manuscript. Users were shown editing results by our method and by the top-3 competing methods (according to the numerical measures), and were asked to choose the preferred result in terms of similarity to the original image, text adherence, and overall. The user-study results show a statistically significant preference to our method over the others.
>
> **Zero-order method for high dimensional data**
>
> This is a great point, as FlowOpt is the first method that performs gradient-free optimization directly over the high-dimensional $\\boldsymbol{z}\_T$ (corresponding to $1024 \\times 1024$ images) and it converges in a small number of iterations, as shown in the paper.
>
> Note that our convergence proof relies on the Banach fixed point theorem [R1], which not only provides a convergence guarantee, but also tells us what the convergence speed is. Specifically, the theorem says that the convergence is geometric, so that in each iteration the distance to the final result is reduced by a constant factor. This factor depends on the contraction constant, which in our case can generally depend on the dimension and the learning rate. But the type of convergence is always geometric, which is rather fast.
>
> To show this, we added Appendix J and Figure S21, where we plot the convergence for images of dimensions $1024 \\times 1024$, $512 \\times 512$, and $256 \times 256$. These graphs show that the shape of the RMSE and PSNR plots tends to look qualitatively similar (the absolute numbers may differ because RMSE on larger images tends to be larger).
>
> [R1] Montesinos, Vicente, Peter Zizler, and Václav Zizler. An introduction to modern analysis. Springer, 2015.‏
>
> **Comparison against gradient-based inversion**
>
> As discussed in the introduction and related-work sections, the only method that uses gradient based optimization over the initial noise is D-Flow, and this method is simply impossible to run on reasonable hardware in reasonable times. This is because, when backpropagating through the entire flow process, the computational graph grows proportionally to the number of flow steps. This imposes extreme memory and compute requirements that render the method unscalable to the models and the resolution of data we used. This is the reason this baseline was omitted from our comparison.
>
> Please note that the original D-Flow paper used a small flow-matching model, small images ($128 \\times 128$), and memory-saving techniques like gradient checkpointing, and still took approximately $15$ minutes to run on a single input. In contrast, here we use large flow models (FLUX and SD3) and large images ($1024 \\times 1024$), which makes D-Flow completely impossible to run.

---

### Official Review · Reviewer_FeKp · 2025-11-01

**Soundness:** 1
**Presentation:** 2
**Contribution:** 1
**Rating:** 2
**Confidence:** 5

**Summary:**

This paper presents FlowOpt, a zero-order (gradient-free) optimization framework for training-free image editing with pretrained diffusion and flow models. Instead of backpropagating through the model or optimizing per timestep.
That said, this work is exactly similar to existing literature, particularly FlowChef, and lacks comprehensive and community-standard evaluations. See details below.

**Strengths:**

* Theorem 1 provides a sufficient condition on the step size under which the FlowOpt iterations provably converge. This formal analysis of convergence is a valuable addition to flow-based optimization literature, where most prior methods rely on heuristic step-size tuning.

**Weaknesses:**

The novelty is limited. The proposed zero-order optimization across the full flow process is conceptually identical to FlowChef [1] (ICCV 2025, arXiv Dec 2024), which already introduced a gradient-free control framework with theoretical guarantees and broad task coverage (inversion, editing, and restoration). The main difference, introducing a step-size bound, is a modest theoretical insight rather than something novel or different.

The work lacks comprehensive evaluation on community-standard editing benchmarks such as PIE-Bench [2], which is now widely adopted for fair comparison across inversion-based and inversion-free methods.

The paper doesn’t clarify the conceptual distinction between FlowOpt and FlowChef, despite their almost identical formulations (both optimize the initial latent by approximating the flow trajectory without backpropagation).

[1] “FlowChef: Steering of Rectified Flow Models for Controlled Generations,” ICCV 2025.
[2] “Direct Inversion: Boosting Diffusion-based Editing with 3 Lines of Code,” ICLR 2024.

**Questions:**

Can the authors clearly articulate the difference between FlowOpt and FlowChef, both theoretically and empirically?

---

> ### Author Response · Authors · 2025-11-20
>
> **Novelty**
>
> We thank the reviewer for pointing out the FlowChef paper, which we mistakenly missed. We added comparisons (theoretical and experimental) that demonstrate the fundamental differences between our approaches.
>
> FlowChef performs optimization on each timestep separately and does not treat the whole flow process as a black-box like our proposed method. Namely, FlowChef optimizes the intermediate latent in each timestep during the sampling process, and not the initial latent as we do. As a result, at the end of the optimization/sampling process, FlowChef does not provide an initial noise vector that reconstructs the original image, which is the goal in image inversion, for example.
>
> More concretely, the equation $z\_t \\leftarrow z\_t - \\eta (\\hat{\\boldsymbol{z}}\_0 - \\boldsymbol{y})$ (step 6 in Algorithm 1 of FlowChef) is a simple guidance step. In contrast, our method does not construct a prediction $\\hat{\\boldsymbol{z}}\_0$ in each timestep $t$ by applying the model to $\\boldsymbol{z}\_t$; it rather constructs $\\boldsymbol{z}\_0$ by applying the whole chain of denoisers to the initial noise $\\boldsymbol{z}\_1$, i.e. $\\boldsymbol{z}\_0 = \\boldsymbol{z}\_1 + \\sum\_{i} \\boldsymbol{v}\_{t\_i}(\\boldsymbol{z}\_{t\_i}) \\, \\Delta t$ (see Equation 3 in our paper). As we illustrate in the revised manuscript, this makes a dramatic difference.
>
> In the updated manuscript, we added a brief explanation about FlowChef in the Related Work section, as well as a more thorough explanation about the inherent differences between FlowChef and FlowOpt in the newly added Appendix K. Moreover, we added FlowChef to the qualitative comparison in Figure 10 and to the quantitative comparisons in Figures 7 and S4. Additional details were added in Appendices B.2.2 and B.2.5.
>
> **Comprehensive evaluation**
>
> Kindly note that the PIE-Bench dataset has become somewhat outdated for benchmarking editing with the new flow models. This is because it contains only small images ($512 \\times 512$) in highly compressed form. Many recent state-of-the-art methods, including FlowEdit, RF-Inversion, FastEdit and Stable-Flow, use different datasets. In particular, some of them construct diverse datasets of high-quality $1024 \\times 1024$ images. Here, we adopted FlowEdit’s published dataset of $1024 \\times 1024$ images and enriched it with additional triplets of the form [image, source prompt, target prompt]. Overall, our dataset contains a highly diverse set of about $400$ triplets.

---

### Author Response · Authors · 2025-12-01
**Final Remarks**

We thank the reviewers for their insightful feedback, we appreciate they found that:
* **The method is novel** and the **presentation is clear** (reviewers NSam and j7YL)
* **The theoretical justification and analysis are novel** (reviewers NSam, j7YL and FeKp)
* **The results are visually appealing and the method succeeds in cases where other methods fail** (reviewers NSam and j7YL)

Below, we highlight key clarifications and revisions addressing the reviewers’ concerns:
* **Theorem assumptions** - we have added a clarification regarding the theorem’s assumptions, and an appendix in which we show that these assumptions indeed hold in reality.
* **Random initialization** - we have conducted an experiment, which was added into the appendix, in which we show that even with random initialization our zero-order optimization method indeed converges, as predicted by our theorem.
* **User study** - we have conducted a user study which confirms the qualitative results - our method is preferred over competing methods.
* **Convergence for other dimensions** - we have added an appendix in which we show that our method converges for other spatial image dimensions as well.
* **Dissimilarity to FlowChef** - we have added a comparison between our algorithm and FlowChef’s algorithm to demonstrate that the methods are fundamentally. Additionally, we have added both quantitative and qualitative comparisons in which we show that our method significantly outperforms FlowChef.

We believe that we have fully addressed the concerns raised, and are hopeful that the AC takes our rebuttal into account, especially with the recent limitations on the discussion period.

---

### Meta-Review · Area_Chair_zcka · 2026-01-07

**Summary:**

In the initial reviews, all three reviewers recommended rejection. Reviewer FeKp appreciated the theoretical analysis, which describes when FlowOpt converges. They complain that the approach is similar to the recent paper FlowChef. In the rebuttal, the authors point out that FlowChef performs optimization on each timestep separately, while the proposed approach treats the entire sampling process as a black box and optimizes only the initial latent (while FlowChef does it per timestep). The reviewer also complains about the lack of benchmarking on PIE-Bench. The authors point to several recent works that (like them) use  higher-resolution datasets. Reviewer j7YL found the idea intuitively appealing, and they appreciated the theoretical contribution and qualitative results. However, they complained about the lack of a user study and asked how the performance of the method changes for high dimensional data. They also ask about comparisons to gradient-based inversion baselines. The authors addressed this by performing a user study and by plotting convergence for different image sizes. They argue that the gradient-based D-Flow method cannot be run on high-resolution images in a reasonable amount of time. Reviewer NSam points out that the method is simple and that it is surprising that it works. However, they complain about the number of NFEs required compared to baselines. They question the validity of the assumptions behind the theorem and point out examples of simple functions for which it would not apply. Finally, they complain about the lack of comparisons to text-based image editing methods and the comparison to Renoise. In the rebuttal, the authors point out that the model obtains strong performance with 100 NFEs (on par with FireFlow and UniInv), as shown in Figure 6. They address the issues with the proof by modifying the theorem's assumptions, to deal with the situation when the supremum is not bounded. The authors also add comparisons to text-based editing methods.

The AC agrees with the reviewers that the method is simple and intuitively appealing. However, after the rebuttal, a number of questions still remain about the scope of the theoretical analysis (by NSam) and to a lesser extent the connection to prior work and evaluation (FeKp, j7YL). The AC therefore feels the paper could benefit from additional revision before acceptance.

**Reviewer Concerns:**

The authors partially address FeKp's concerns by clarifying the differences between FlowChef and this method, and pointing out limitations in the proposed benchmark. The authors addressed reviewer j7YL's concerns about evaluation through a user study. The rebuttal contains significant changes to the assumptions behind the theorem to address NSam's concerns.

**Reviewer Scores:**

It is unclear whether the proposed changes to the theoretical analysis (including the adjusted assumptions) fully address NSam's concerns, so it is hard to predict whether they would adjust their score. Reviewer FeKp may have improved their score (from Reject) to a higher score, given the explanation of the differences to related work, though it is difficult to predict if this would have been satisfactory for them. The user study likely at least partially addresses j7YL's concerns, so they may have increased their score.

---

### Decision · Program_Chairs · 2026-01-26

Reject